# Predicting Tissue Loads in Running from Inertial Measurement Units

**DOI:** 10.3390/s23249836

**Published:** 2023-12-15

**Authors:** John Rasmussen, Sebastian Skejø, Rasmus Plenge Waagepetersen

**Affiliations:** 1Department of Materials and Production, Aalborg University, Fibigerstraede 16, 9220 Aalborg East, Denmark; 2Department of Public Health, Aarhus University, Bartholins Allé 2, 8000 Aarhus, Denmark; sdsk@ph.au.dk; 3Research Unit for General Practice, Aarhus University, Bartholins Allé 2, 8000 Aarhus, Denmark; 4Department of Mathematical Sciences, Aalborg University, Skjernvej 4A, 9220 Aalborg East, Denmark; rw@math.aau.dk

**Keywords:** running, injuries, Achilles tendon, patella ligament, IMU, data science, biomechanics, public health

## Abstract

Background: Runners have high incidence of repetitive load injuries, and habitual runners often use smartwatches with embedded IMU sensors to track their performance and training. If accelerometer information from such IMUs can provide information about individual tissue loads, then running watches may be used to prevent injuries. Methods: We investigate a combined physics-based simulation and data-based method. A total of 285 running trials from 76 real runners are subjected to physics-based simulation to recover forces in the Achilles tendon and patella ligament, and the collected data are used to train and test a data-based model using elastic net and gradient boosting methods. Results: Correlations of up to 0.95 and 0.71 for the patella ligament and Achilles tendon forces, respectively, are obtained, but no single best predictive algorithm can be identified. Conclusions: Prediction of tissues loads based on body-mounted IMUs appears promising but requires further investigation before deployment as a general option for users of running watches to reduce running-related injuries.

## 1. Introduction

Running is a popular physical activity not only for recreational purposes, but also among elite athletes. For instance, a survey covering the years 2015 through 2022 in England [1] showed that about 10% of the population, corresponding to roughly 6 million people, regularly engage in running. Running thus engages enough participants to have an impact on the overall activity level of the population, and the health benefits of active lifestyles are well documented [2,3].

Unfortunately, running is also associated with a high risk of sustaining a running-related injury, and injury incidence between 8 and 18 injuries per 1000 h of running has been reported [4]. Especially the knee and the ankle are susceptible to injuries with Achilles tendinopathy being the most incident injury and patellofemoral pain syndrome being the most prevalent injury [5]. Apart from the immediate discomfort and long recovery time associated with running-related injuries [6,7], injuries are also the most common reason for stopping running [8,9]. Therefore, it is imperative to mitigate the risk of running-related injuries, which requires a profound understanding of why these injuries occur [10].

Running-related injuries are commonly assumed to be caused by the repetitive loading of the affected tissues, resulting in inflammation and tissue failure over time [11]. Loads on muscles, tendons, and bones are, however, difficult—bordering on impossible—to measure in vivo, even in advanced laboratories [12]. Advances in model-based simulation of musculoskeletal forces over the past two decades have provided algorithms [13,14] and models [15] that verifiably predict internal musculoskeletal forces, but these methods rely on complete kinematic input, typically obtained from optical motion capture systems [16] or motion capture suits [17]. Optical systems are impractical in field settings, and both types are too comprehensive for research with large cohorts as well as to inform the individual runner. Therefore, there is a need for light-weight technologies that can accurately estimate tissue loads and injury risk factors in field settings.

Such technology could be running watches, which many recreational and elite runners rely on to track their training efforts and performance [18,19], and whose popularity has also been attributed to independence from organized coaching [20]. Running watches typically record biometrics, such as pulse, and kinematic data, such as accelerations and GPS positions, from which running speed, step frequency and step length can be derived.

Recent developments in data science have inspired data-based models linking inertial measurement unit (IMU) kinematics, complete kinematics [21] and musculoskeletal kinetics [22,23,24,25,26,27,28]. Such models can be trained on musculoskeletal simulations based on optical motion capture data and have shown promising results for modeling of a variety of tasks, but on relatively small and uniform cohorts of test subjects.

The data space of all possible movements performed by all possible people is very large, and the connection between kinematics and kinetics expressed by the laws of dynamics is highly nonlinear, so it is unlikely that machine learning will outcompete physics-based models in general. However, running is a small subset of human movements, and given runners’ need for fast and lightweight feedback from IMUs, it is worth investigating whether reliable kinetic estimations, i.e., tissue loads, are possible from the combination of IMU data, a database of verified running kinematics for a medium-size heterogeneous population, and simple anthropometrics for the individual runner.

When choosing predictors, there are a few considerations to keep in mind. First and foremost, the predictors should be feasible and inexpensive to collect in field-based settings and, therefore, require as little additional equipment as possible. Secondly, to be relevant for performance and injury mechanisms, the predictors should carry as much information as possible about running biomechanics. Thirdly, the total of number of predictors should be curbed to avoid overfitting.

IMUs, such as those embedded in running watches and smartphones, are inexpensive and feasible to use in field-based settings. They provide kinematic data only for their attachment points, but they typically sample at frequencies of 400–500 Hz [29] and therefore accumulate a large amount of data over a short time. In the interest of data reduction, some studies extract a few features from the raw signal’s time domain, such as the maximum angle of a segment or peak accelerations, which capture some, but not all, aspects of running kinematics. An alternative approach is to transform the entire signal to the frequency domain using discrete Fourier transformation. Given the periodic nature of the data, this provides an accurate, yet compressed, description of the entire running kinematics [30]. We investigate the prediction of Achilles tendon and patella ligament loads based on data from IMUs positioned on easily accessible anatomical locations, i.e., wrists, ankles, and the sternum, and we compare IMU positions and sets of predictors to optimize the results.

## 2. Materials and Methods

Figure 1 illustrates the data flow and computational investigations.

### 2.1. Experimental Data

The experimental and biomechanical simulation procedures were described previously in detail [30], and are briefly summarized here for completeness: a nine-camera Qualisys Miqus system (Qualisys AB, Gothenburg, Sweden) was used to collect full-body optical marker data for treadmill running for 78 runners (30 female, 48 male) in 285 trials (180 male trials and 105 female trials) with speeds ranging between 6 and 20 km/h at 300 frames per second and a resolution of 2 megapixels. The subjects were asymptomatic and ranged in skill level from beginner to elite. The optical marker data were transferred to a physics-based simulation in the AnyBody Modeling System version 7.2 (AnyBody Technology A/S, Aalborg, Denmark) [13], which converted the marker data to anthropometrical data, i.e., individual segment dimensions, and anatomical joint angle time series [31]. Using the Twente Lower Extremity Model version 2.0 [32], this process also simulates internal biomechanical kinetics. For the purposes of this paper, simulated maximum values of the patella ligament force and the Achilles tendon force over the running cycle were stored for each recorded trial and normalized by body mass.

The time series of anatomical joint angles were segmented into strides, which were transferred to the frequency domain by Fast Fourier Transform (FFT), retaining five sine and five cosine terms and the DC (constant) component, i.e., 11 coefficients to describe the motion of each of the 88 independent anatomical degrees-of-freedom. This procedure was described in detail previously [30].

Using the model, we emulated five virtual, three-axis accelerometers located on the two wrists, the two ankles and the sternum. The accelerometers were constructed in the model as body segment-fixed local reference frames, whose acceleration vectors could be extracted. The placements and local coordinate systems are illustrated on Figure 2. Gyroscopic information was not included. The virtual accelerations were combined, after the appropriate coordinate transform, with the gravity component to mimic the accelerations including gravity that would have been measured by body-fixed IMUs. Time series for the resulting virtual accelerations in segment-fixed *x*, *y,* and *z* coordinates along with the magnitude of the acceleration vectors were also subjected to FFT transformation, retaining 11 Fourier coefficients for each direction and the magnitude, leading to a total of 44 coefficients for each accelerometer.

This study used a previously anonymized version of the data resulting from the data processing described above. The process results in a matrix with 285 rows corresponding to the recorded trials. The number of columns is 1229, of which 968 contain Fourier coefficients for the model’s kinematic degrees-of-freedom, and 220 = 5 × 44 columns contain Fourier coefficients for the virtual accelerations of the five selected placements. The maximal forces over the cycle for the right and left patella ligaments and Achilles tendons were extracted from the kinetic analysis, yielding four additional columns. Furthermore, 35 columns contain anthropometric measurements such as segment lengths, body weight, stature, and gender. Finally, the table contains running speed and angular step frequency.

Initial investigations of virtual accelerometer data revealed a high degree of symmetry between the left and right ankles. It was therefore decided to discard the right ankle and include only accelerations of the two wrists, the left ankle, and the sternum, in total 176 columns of Fourier coefficients of acceleration data. It was furthermore decided that combinations of more than two IMUs would be impractical for most runners and, consequently, the following combinations were selected for further investigation:left wristleft wrist and right wristleft wrist and sternumleft wrist and left anklesternum and left ankleleft anklesternum.

Single-IMU options 1, 6 and 7 lead to inclusion of 1 × 4 × 11 = 44 columns of Fourier coefficients, and double-IMU options 2–5 each lead to inclusion of 88 columns. In the interest of simplicity of the procedure for the runner and to minimize the risk of overfitting, only the step frequency, sex, age, thigh length, shank length, foot length, body mass, stature, BMI and running speed were included as additional variables, leading to 54 or 98 predictor variables for single and double-IMU configurations, respectively.

### 2.2. Prediction Algorithms

We are in a setting with a relatively low number of observations (285 running trials) compared to the number of predictor variables (54 or 98, depending on whether a single-IMU or double-IMU configuration is used). A standard multivariate linear regression model is therefore not suitable since it will be prone to overfitting. Thus, we consider regularized versions of multivariate linear regression using the elastic net approach [33], which encompasses ridge regression, LASSO (Least Absolute Shrinkage and Selection Operator), and combinations of ridge regression and LASSO. The elastic net further enables variable selection that discards weak predictors. This has the potential to reduce the number of variables needed for prediction, which can make the method more feasible for future users.

A regularized linear regression is ”easy” to interpret, but clearly has shortcomings in case of nonlinear relationships or interaction effects (unless the latter are explicitly included in the model). Therefore, we also consider a flexible tree-based machine learning method based on gradient boosting using specifically the computationally efficient XGB algorithm [34]. In this case, the prediction is obtained from a sequence of relatively shallow trees. These trees are fitted sequentially, where each new tree is fitted to residuals arising from prediction by the current sequence of trees. The splits employed when constructing prediction trees are very useful for handling nonlinearities and interactions.

#### 2.2.1. Tuning Parameters

The elastic net approach relies on two tuning parameters, α and λ. The first of them, α, ranges between 0 and 1, where 0 gives ridge regression, 1 gives LASSO, and intermediate values provide combinations of ridge and LASSO. For α, we consider the values 0, 0.5 and 1. The other parameter, λ, determines the strength of regularization and is determined by minimizing a cross validation score. Following the implementation in the glmnet package [35], the data is partitioned into 10 subsets. For each subset, the model is trained on the remaining data and is then used to predict the observations in the subset. The resulting mean square prediction errors over each subset are averaged to get the cross-validation score.

For XGB, there is a wide range of tuning parameters. Here, we restrict attention to the important number of sequentially fitted trees *m*, tree depth *d* and learning rate η and leave the remaining tuning parameters at their default values. The number of trees *m* is chosen by cross-validation as implemented in the XGB package (with 5 subsets, referring to the explanation above of cross-validation). Following recommendations in the machine learning literature, we further consider rather shallow trees, *d* = 3, 6 and η = 0.1, 0.3. There is a trade-off between the latter two parameters, so that deeper trees should in general be combined with a lower learning rate and vice versa.

#### 2.2.2. Model Evaluation and Selection

The models with varying values of tuning parameters are trained on a random subset of the data containing 90% of the observations (the training set) and the prediction performance is next evaluated on the remaining 10% of the observations (the test set). To avoid sensitivity to the random splitting into training and test data sets, we consider 2500 independent random splits (with replacement), carry out the training and test procedure on each resulting training/test data set, and average test results over the 2500 splits. We measure prediction performance in terms of average correlation and normalized root mean square error (nRMSE). For each test set, the correlation is between the physics-based values of the dependent variables and the predicted values obtained from the model trained on the training data set. The root mean square error (RMSE) is the square root of the average of squared differences between physics-based values in the test data set and corresponding predicted values. The RMSE is normalized by dividing by the average of the dependent variable over the full data set. In a few cases, the elastic net method produced a constant predictor. Then, the correlation is not well-defined due to division by zero. We hence ultimately use nRMSE for identifying the best models.

## 3. Results

### 3.1. Comparison of Prediction Methods and Configurations of Accelerometers

Figure 3 and Figure 4 show performance measures (correlation and normalized RMSE) when patella ligament forces and Achilles tendon forces, respectively, are predicted using elastic net and XGB with different settings of tuning parameters and configurations of accelerometers.

Considering patella ligament force, best prediction results were obtained when at least one measurement unit was placed at the ankle. In that case, elastic net is superior to XGB and correlations up to 0.96 are achieved. For the Achilles tendon force, XGB generally outperforms elastic net, and the best results were obtained with at least one measurement unit placed at the sternum. For the Achilles tendon force, correlations up to 0.72 were achieved. Generally, both for the patella ligament force and the Achilles tendon force, and over the seven configurations of accelerometers, alpha equal to 0.5 or 1 (elastic net) and shallow trees of depth 3 (XGB) gave the best results in terms of nRMSE.

For the patella ligament force and Achilles tendon force alike, the best results were obtained when just one measurement unit was used (at the ankle for the patella ligament force and at the sternum for Achilles tendon force).

### 3.2. Detailed Inspection of Best Prediction Methods and Measurement Unit Configurations

To closer inspect the prediction methodology, we considered the best configurations of prediction algorithms and accelerometer configurations, i.e., elastic net with one measurement unit at the ankle and α = 1 for the patella ligament force, and XGB with one unit at the sternum, *d* = 3 and η = 0.3 for the Achilles tendon force. Figure 5 and Figure 6 show scatter plots (so-called calibration plots) of physics-based values and predictions from the test data sets. To avoid too dense scatterplots, we only included 250 randomly sampled points from the test data sets. The solid and dashed lines are the identity and the least squares lines, respectively. The normalized RMSE and correlation are 0.12 and 0.95, respectively for the patella ligament scatter plot and 0.18 and 0.71 for the Achilles tendon scatter plot. A moderate bias in the predictions for the Achilles tendon force is revealed by the discrepancy between the identity and least squares lines, whereas bias is essentially absent for patella ligament force predictions.

### 3.3. Variable Importance

The importance of the various predictor values was assessed for the patella ligament force with elastic net and one measurement unit at the ankle. With α = 1, elastic net may estimate some predictor coefficients to be exactly zero, so that the corresponding predictor has no effect on the prediction. Thirteen variables were included in less than 10% of the test/training data sets. These are age, body mass, stature and ten acceleration measurement coefficients. Nineteen variables were included in at least 90% of the cases. These are the angular frequency, gender, shank length and 16 acceleration coefficients. The predictors can also be ranked according to their estimated effect sizes. The 10 variables with highest average rank over test/training data sets were (in rank order):the average acceleration magnitude of the left anklethe second sine coefficient of the acceleration magnitude of the left anklethe first cosine coefficient of the acceleration magnitude of the left ankleshank lengththe first sine coefficient of the acceleration magnitude of the left anklethe third cosine coefficient of the anterior/posterior acceleration of the left anklethe first sine coefficient of the anterior/posterior acceleration of the left anklethe third sine coefficient of the acceleration magnitude of the left anklethe fourth cosine coefficient of the acceleration magnitude of the left anklethe second sine coefficient of the anterior/posterior acceleration of the left ankle

The angular frequency of the Fourier series and gender have ranks 15 and 22, respectively. Overall, the data do not support reducing the number of predictors for the patella ligament force significantly.

In case of the XGB predictions for the Achilles tendon force using sternum acceleration measurements, the importance of variables may be assessed by their so-called gain, which measures the contribution of a variable to the prediction. We assessed the average gains and average ranks of variables according to their gains over the 2500 test/training data sets. The ten variables with the highest average gains are:The fourth sine coefficient of the vertical acceleration of the sternum;The third sine coefficient of the lateral acceleration of the sternum;BMI;The third cosine coefficient of the lateral acceleration of the sternum;The second sine coefficient of the anterior/posterior sternum acceleration;Shank length;The second sine coefficient of the magnitude of the sternum acceleration;Foot length;The first cosine coefficient of the anterior/posterior sternum acceleration;The second sine coefficient of the vertical acceleration of the sternum.

It is worth noticing that the gains taper off rapidly, with variables from rank 8 and upwards sharing similar, small gains with variables outside the list. The variables speed and gender have lowest and third lowest average ranks, respectively. The remaining additional (non-acceleration) variables are among the 50% variables with the highest rank; except perhaps for speed and gender, there does not seem to be an obvious opportunity to eliminate additional variables.

## 4. Discussion

Running can be performed at a wide range of speeds, and runners exhibit different styles depending on their anthropometry and other physiological preconditions. The participants in the present study are all able-bodied runners, but the data represent speeds between 6 and 20 km/h and skill levels between beginner and elite, and the resulting biomechanical loads vary considerably. Nonetheless, despite the complexities of the data, we obtain satisfactory correlations and prediction errors, and the presented method is a promising approach to predicting running biomechanics in field settings.

The present study differs from most previously reported results by targeting so-called structure-specific loads, i.e., the force on the Achilles tendon and patella ligament. To our knowledge, only one other study has attempted to predict these structure-specific loads from wearable devices [36]. In this study, Brund et al. found mean absolute percentage errors ranging from 13 to 30%, which appear similar to the normalized root mean squares error of 15–20% we report for the best performing models. However, the predictions by Brund et al. show markedly worse calibration upon visual inspection, which may be a result of the much lower number of predictors (five) and less sophisticated prediction models (simple multiple linear regression) compared to the present study.

A couple of studies have predicted tibial bone forces from wearable devices and found accurate predictions with mean absolute percentage errors ranging from 2.6 to 17.9% [28,37]. However, these studies are dependent on measurements from pressure-sensing insoles, which are not common equipment for runners currently. Most other prediction studies using wearable devices have not predicted structure-specific loads but rather net joint moments, ground reaction forces (and derivatives thereof), or kinematics such as joint angles or stride length [38]. While such measures might be interesting in other contexts, an injury prevention context begs for structure-specific loads as these are closest to the actual injury mechanism [39].

In terms of correlations, the patella ligament loads are clearly better predicted than Achilles tendon loads for all IMU configurations. Our predictions are further inferior to those reported by Long et al. [40], who, however, considered a much smaller and more homogeneous cohort of four male basketball players. The Patella ligament force is strongly related to quadriceps activity, which again is related to the exerted knee extension moment. The latter is given by the product of the vertical acceleration of the masses above the knee and the moment arm, which is given by the knee flexion. IMU data reflecting these properties would convey the necessary information to accurately assess patella ligament loads, as shown by the results.

Prediction of the Achilles tendon force, on the other hand, appears to be more challenging. The explanation might be found in the binary nature of the foot contact mechanism, where small motion differences determine contact or non-contact of specific parts of the foot. Indeed, forefoot versus heel strike running styles cause different ground reaction force patterns and different load patterns of the Achilles tendon, even though the kinematic differences in terms of heel position can be quite small. Possibly for this reason, correlations for the Achilles tendon force peak around 0.70. It is possible that inclusion of a categorical variable for heel versus forefoot strike would improve the predictions.

The normalized root mean square errors for both output variables, i.e., patella ligament force and Achilles tendon force, appear to be more similar than the correlations, with best results for both cases in the range of 15–20%. The non-negligible root mean square errors mean that it is commendable to present, e.g., 95% prediction intervals to users rather than just one number. Correlations, on the other hand, do not contain information about the precision of the absolute numbers but rather about the ability of the model to capture changes correctly. This can be useful for an individual runner who considers a change in running style and wants to know whether that would influence the load on a given tissue positively or negatively.

The ten first predictors in terms of rank contain several sine and cosine terms of higher order, which might be surprising given that the higher-order Fourier coefficients are supposed to diminish. However, a running cycle comprises the time from right heel strike to right heel strike, which causes the leg and arm movements to have their dominant frequency at the step cycle frequency, while torso motions tend to repeat with double frequency, where terms from second order and upwards dominate.

The results do not support identification of a single best type of algorithm. Elastic net performs better than XGB for patella ligament force and vice versa for Achilles tendon force. However, XGB may have some merit over elastic net, in the sense that prediction results for XGB are more stable over IMU placement while, especially for Achilles tendon force, elastic net results are quite sensitive to IMU placement. For both algorithms and forces to be predicted, using a single IMU gave better results than using two. The reason may be that the resulting lower number of Fourier acceleration coefficients (44 rather than 88) guards against overfitting.

The high correlations for especially the patella ligament force suggest that the method has the potential to evolve into a reliable information source for injury prevention efforts. Speculatively, our method could be used to estimate whether a change in running biomechanics would lead to changes in injury risk for a given runner. However, running biomechanics is complex, and an attempt to offload a given anatomical structure might increase the load of other structures. Running habits, such as speed and weekly distance, might also be affected by a person’s running biomechanics and affect injury risk indirectly.

In this study, the two structure-specific loads were included as their respective maximum values over the running cycle. These values may occur at different times in the running cycle depending on the running style, for instance heel or forefoot strike. Future studies may consider predicting the entire cycle of the forces as a more informative measure of the load exposure of the tissues.

It should be noted that the accelerations, from which the loads are predicted, are perfect virtual accelerations from treadmill running that are not affected by measurement noise, soft tissue artefacts, uncertain positioning, and variable terrain circumstances, which will be confounding factors for real measurements. A similar approach was previously employed [28,37] by first developing a tibial bone force prediction model using virtual instruments in [28] and subsequently validating the prediction model using physical instruments in [37].

## 5. Conclusions

Data-based methods for predicting structure loads in running from a small number of accelerometers appear to have merit in the case of running. However, further investigations of data-based methods against physics-based models and in vivo experimental data, including injury registration, and the inclusion of accelerations measured in field conditions, are commendable.

## Figures and Tables

**Figure 1 sensors-23-09836-f001:**
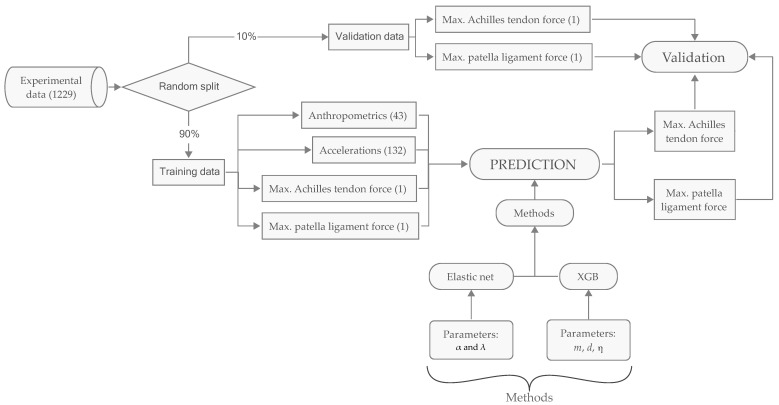
Data flow and computational methods. Figures in parentheses designate numbers of variables. The collected data set contains a total of 1229 variables of which 43 are virtual accelerometer data and 132 are anthropometric parameters. The prediction methods are, respectively, Elastic Net with two tuning parameters and XGB with three tuning parameters. The experimental data are randomly split into training and validation set.

**Figure 2 sensors-23-09836-f002:**
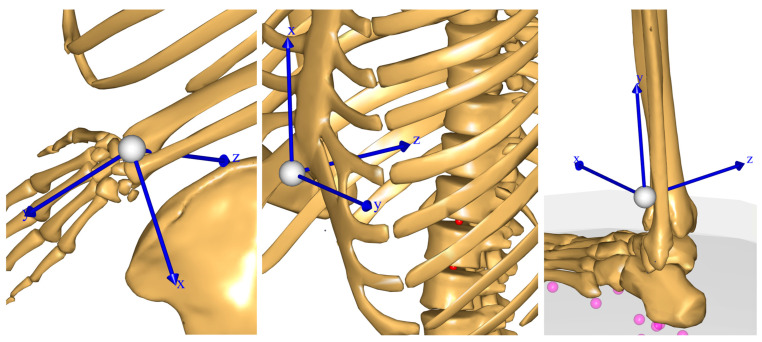
Placement of virtual IMUs on the left wrist, sternum and left ankle, and their respective local coordinate systems. The IMU on the right wrist is placed similarly to the IMU on the left wrist. The purple dots are floor contact points, and the red dots are spinal joint centers.

**Figure 3 sensors-23-09836-f003:**
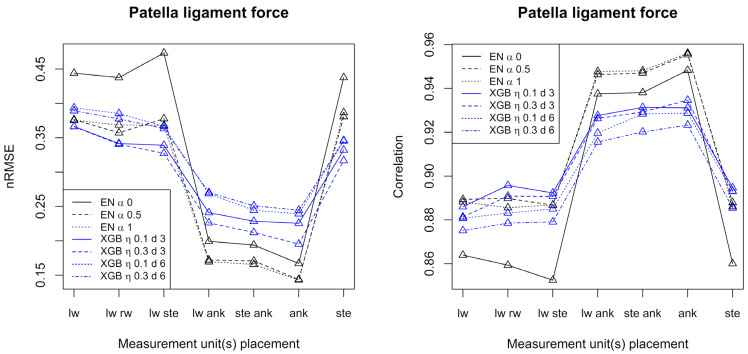
Performance measures for prediction of patella ligament forces by elastic net and XGB, respectively. The left frame shows the normalized root mean square error, and the right frame shows the correlation. Conditions: lw = left wrist, rw = right wrist, ste = sternum, ank = ankle.

**Figure 4 sensors-23-09836-f004:**
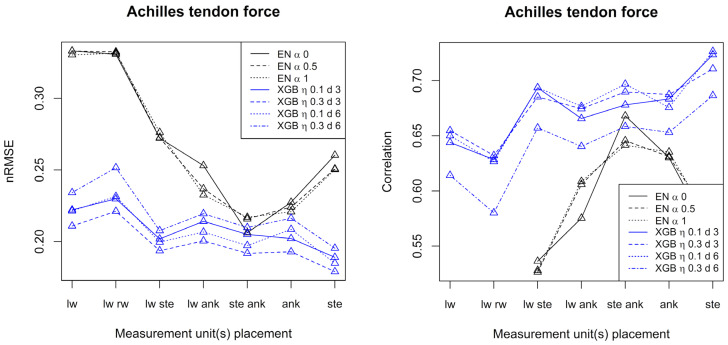
Performance measures for prediction of Achilles tendon forces by elastic net and XGB, respectively. The left frame shows the normalized root mean square error, and the right frame shows the correlation. Conditions: lw = left wrist, rw = right wrist, ste = sternum, ank = ankle.

**Figure 5 sensors-23-09836-f005:**
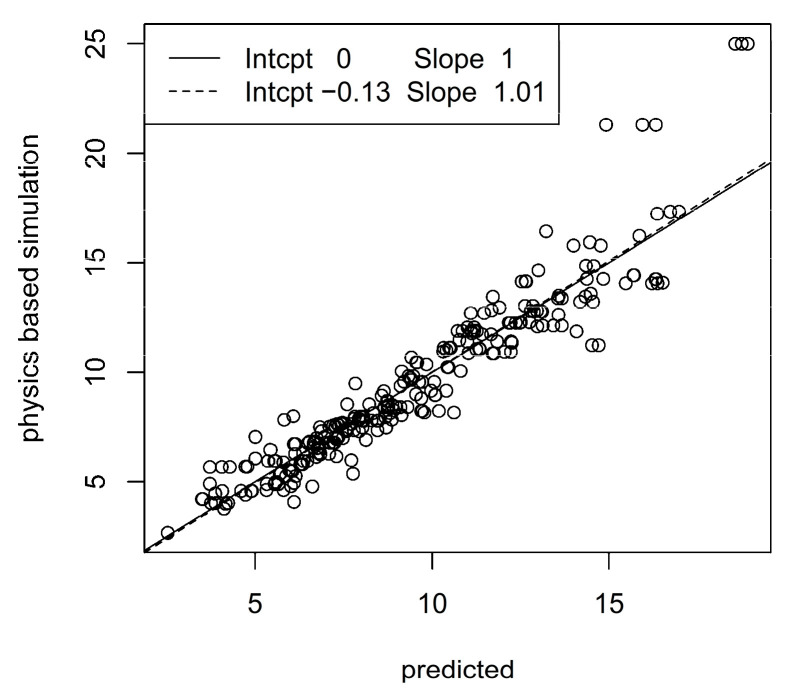
Correlation between predicted and physics-based simulated patella ligament forces. The legend shows intercepts and slopes of identity line (solid) and least squares line (dashed).

**Figure 6 sensors-23-09836-f006:**
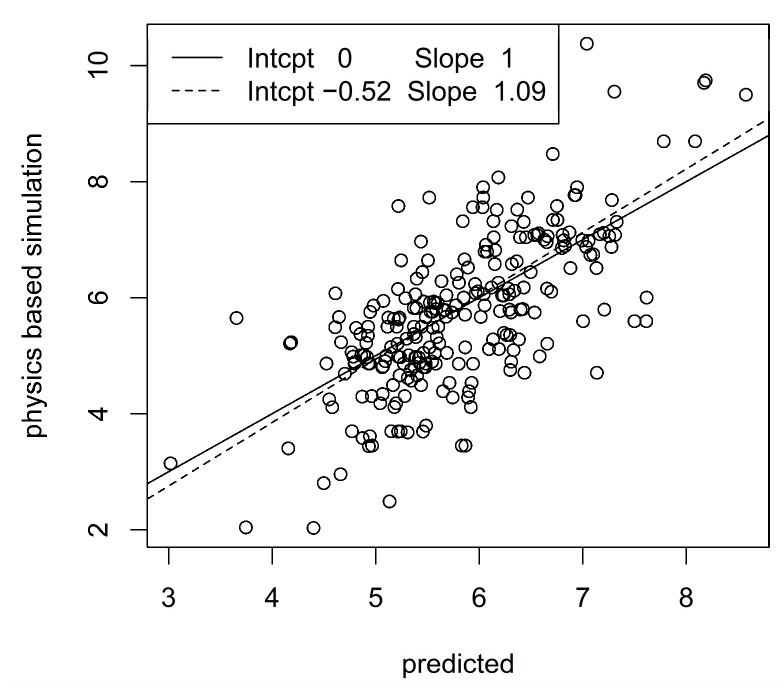
Correlation between predicted and physics-based simulated Achilles tendon forces. The legend shows intercepts and slopes of identity line (solid) and least squares line (dashed).

## Data Availability

Anonymized data can be made available by personal contact to the authors.

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
