# Peer review of "Predicting Tissue Loads in Running from Inertial Measurement Units"

_sensors, 2023, doi:10.3390/s23249836_

Round 1
Reviewer 1 Report
Comments and Suggestions for Authors
Overall I thought this was an interesting manuscript. I have some minor comments listed below.
1) Line 109: There is a typo of "intro" instead of "into". Please review the manuscript for additional typos/minor grammatical errors.
2) Figures 3 and 4 are a bit difficult to visualize. I would suggest making them bigger so that the distinction between each line is easier to see.
3) The authors state that they assessed prediction of patellar and achilles tendon forces from right heel strike to right heel strike, which includes both a contact and swing phase.
I would assume that patellar and Achilles tendon forces are much smaller during swing, compared to when the foot is in contact with the ground. Does including the swing phase potentially add a larger amount of variability to the overall model estimations and in turn reduce the overall correlations particularly for the Achilles tendon force?
Comments on the Quality of English LanguageOverall, the paper is well written and the quality of English is good. As mentioned, there is a typo on line 109 and the authors should review the paper for additional typos, etc.
Reviewer 2 Report
Comments and Suggestions for Authors
Authors explore a hybrid approach that combines physics-based simulations and data-driven techniques. Authors utilized 285 running trials conducted by 76 real runners to perform physics-based simulations, enabling the retrieval of forces acting on the Achilles tendon and patella ligament. Subsequently, the gathered data is employed for both training and testing a data-driven model, utilizing elastic net and gradient boosting methods.
My specific comments are as follow:
- Figure 1 should be more described by explaining the used approach. In addition, data flow should be mentioned illustrating the inputs/outputs of each block.
- Experimental platform should be more detailed. Authors can add a real figure of the real experimentation illustrating how the data were acquired, stored and processed.
- Line 95: Please include more details about the experimental and biomechanical simulation procedures. Indeed, this part lacks some details regarding the selection of the studied speed range, technical specifications of the used hardwares (camera), parameters of the physics-based simulation software, size of the generated data.
- line 115: How the acceleration vectors were extracted ? and what is the size of the generated data ?
Especially for data processing, are there specific algorithms dedicated to eliminating interference and drifting? In addition, why was the gyroscopic information not included ? Authors should specify the technical specifications of the used IMU, this is very important to estimate the reliability of the generated data. The acquisition parameters should be mentioned, especially for the sampling frequency.
- Line 119: the transformation of the time domain data to the frequency domain should be more illustrated.
- Figure 2: Authors should specify why the IMU on the right wrist is not used.
- Line 128: Authors should specify what is the novelty against their previous work.
-Line 167: Authors claim "A regularized linear regression is easy to interpret but obviously has shortcomings in case of nonlinear relationships or interaction effects". How the nonlinearity issues are resolved by the proposed approach?
- Section 2.2: it's unclear how the prediction algorithms work. How are the tuning parameters selected ?
- The Conclusion should be added by integrating the limitations and the perspective.
Round 2
Reviewer 2 Report
Comments and Suggestions for Authors
After thorough revision and enhancement, this paper now demonstrates significantly improved clarity, depth, and coherence in presenting its findings and arguments.